# Positive and negative regulation of carbon nanotube catalysts through encapsulation within macrocycles

Matías Blanco[1], Belén Nieto-Ortega[1], Alberto de Juan[1], Mariano Vera-Hidalgo[1], Alejandro López-Moreno [1], Santiago Casado[1], Luisa R. González[2], Hidetaka Sawada[3], José M. González-Calbet[2] & Emilio M. Pérez [1]

One of the most attractive applications of carbon nanomaterials is as catalysts, due to their extreme surface-to-volume ratio. The substitution of C with heteroatoms (typically B and N as p- and n-dopants) has been explored to enhance their catalytic activity. Here we show that encapsulation within weakly doping macrocycles can be used to modify the catalytic properties of the nanotubes towards the reduction of nitroarenes, either enhancing it (n-doping) or slowing it down (p-doping). This artificial regulation strategy presents a unique combination of features found in the natural regulation of enzymes: binding of the effectors (the macrocycles) is noncovalent, yet stable thanks to the mechanical link, and their effect is remote, but not allosteric, since it does not affect the structure of the active site. By careful design of the macrocycles' structure, we expect that this strategy will contribute to overcome the major hurdles in SWNT-based catalysts: activity, aggregation, and specificity.

[1] IMDEA Nanociencia, Ciudad Universitaria de Cantoblanco, c/Faraday 9, 28049 Madrid, Spain. [2] Departamento de Química Inorgánica, Universidad Complutense de Madrid, 28040 Madrid, Spain. [3] JEOL Ltd, 3-1-2 Musashino, Akishima, Tokyo 196-8558, Japan. Correspondence and requests for materials should be addressed to E.M.Pér. (email: emilio.perez@imdea.org)

Nature uses a variety of strategies to regulate the activity of enzymes. Mechanisms include complex multimolecular approaches, such as compartmentalization within specific organelles or increasing the concentration of enzyme within a protein scaffold, but the most general methods imply the supramolecular or covalent modification of the enzyme's structure. Direct competition for the active site is the simplest supramolecular regulatory mechanism. Allosteric regulation implies a conformational change in the three-dimensional (3D) structure of the enzyme's active site in response to the noncovalent binding of an effector to a regulatory site located far from it. Phosphorylation, the hydrolysis of GTP to GDP by GTP-binding proteins, and (poly)ubiquitination are the most general methods of regulation based on the making and breaking of covalent bonds[1].

Metal-free catalysis is one of the most attractive applications of carbon nanomaterials[2–5]. However, the mechanisms explored to regulate their catalytic activity are limited to positive regulation via covalent modification of their native structure, mostly by including heteroatoms[6–9].

We, and subsequently Miki et al., have recently reported strategies to form rotaxane-like mechanically interlocked nanotube (MINT) derivatives[10,11]. The native structure of single-walled carbon nanotubes (SWNTs) is preserved upon formation of MINTs, while the addition of the macrocycles can prevent bundling at the nanoscale[12–15]. We have shown that these unique features make MINTs superior polymer fillers[16] and are in principle very appealing for their application in catalysis. The carbon surface of nanomaterials acts as both adsorbent and facilitator of the electron-transfer process in nitroarene

reductions[17], while the modification of the electronic properties of SWNTs through supramolecular modification with electroactive molecules is a well-documented phenomenon[18,19]. Moreover, mechanically interlocked molecules have shown distinctive advantages in the regulation of catalytic activity[20–22]. Based on these facts, we expected that the catalytic activity of SWNTs would be controlled remotely by encapsulation with suitable n- or p-doping macrocycles. Here we show that encapsulation of SWNTs within electroactive macrocycles to form MINTs is a valid strategy for the positive and negative regulation of the catalytic activity of SWNTs.

## Results

**Synthesis and characterization.** We used (6,5)-enriched SWNTs ((6,5)-SWNTs; 0.7–0.9 nm in diameter, length >700 nm, 95% purity) in all experiments. The structures of all macrocycles and density functional theory (DFT) optimized geometries of MINTs with (6,5)-SWNTs are shown in Fig. 1a, b, respectively. Mac-exTTF and mac-pyr and their corresponding MINTs were synthesized as previously reported and fully characterized using thermogravimetric analysis (TGA), Raman spectroscopy, ultraviolet–visible–near infrared absorption spectroscopy (UV-Vis-NIR), photoluminescence excitation/emission spectroscopy (PLE) maps, high-resolution transmission electron (HR-TEM), and atomic force microscopies (AFM) and, most importantly, adequate control experiments[14,15].

Mac-AQ shows identical structure to mac-exTTF but features the well-known electron acceptor AQ[23] (see Supplementary

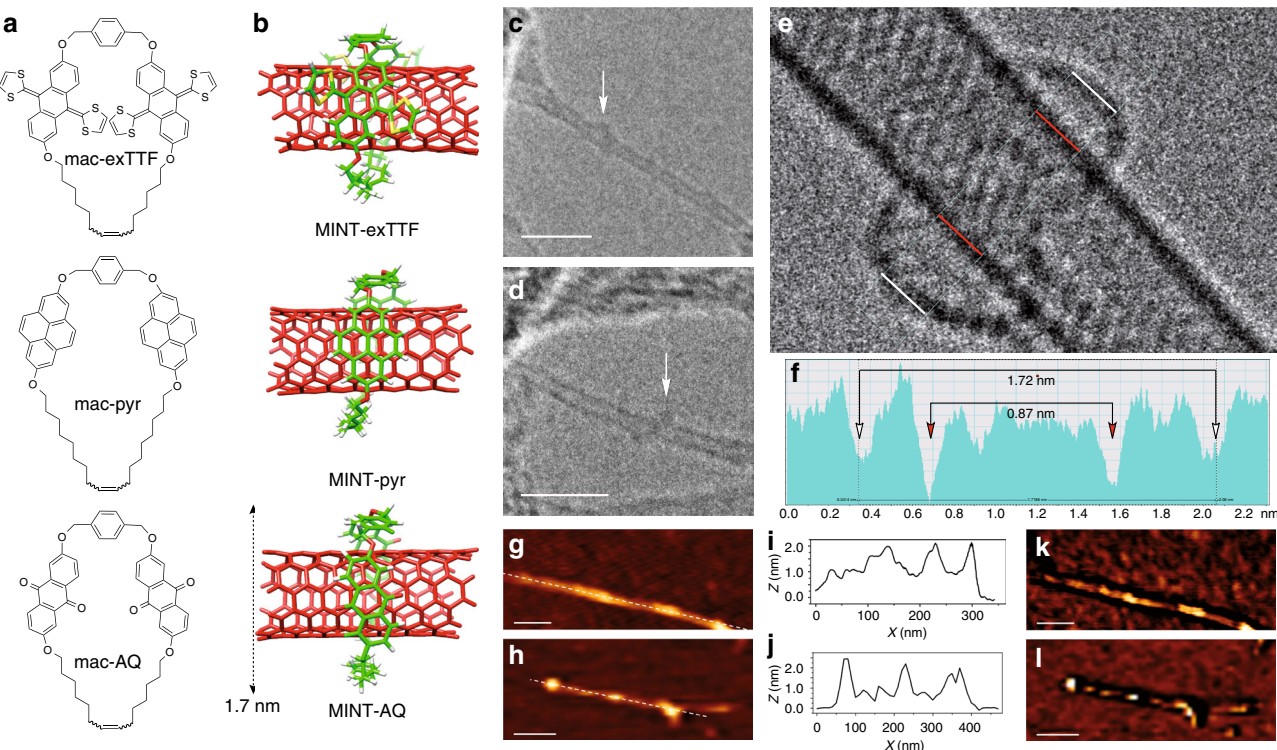

**Fig. 1** Structures and microscopy. **a** Chemical structure of the macrocycles and **b** minimum energy (DFT) geometries of the corresponding MINTs with (6,5)-SWNTs. The calculated diameter of MINT-AQ is shown for comparison with the microscopy images. **c, d** Representative TEM images of MINT-AQ, showing SWNTs surrounded objects of adequate size (ca. 2 nm) and shape to be mac-AQ. Scale bars are 5 nm. **e** ac-HRTEM image of a single MINT-AQ, and **f** its corresponding analysis along the box depicted with thin white and red lines. **g–l** AFM characterization of MINT-AQ. We observe isolated SWNTs with protrusions of around 2 nm (**g, h**), as shown in the profiles along the white dashed lines (**i, j**). The phase images (**k, l**) show energy dissipation contrast at the protrusions. Scale bars are 50 nm (**g, k**) and 100 nm (**h, l**)

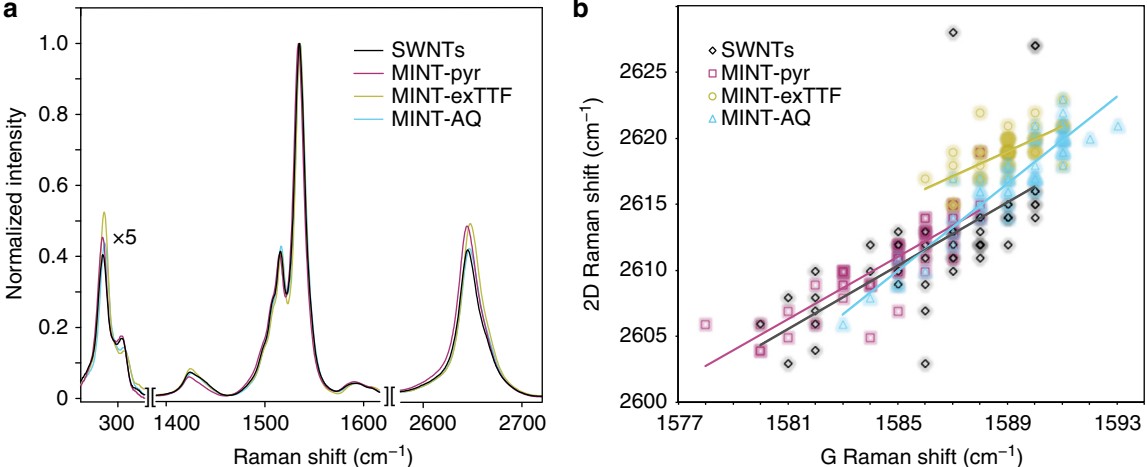

**Fig. 2** Raman spectroscopy. **a** Average ($N = 50$) Raman spectra of SWNT (black), MINT-exTTF (yellow), MINT-pyr (purple), and MINT-AQ (cyan). **b** Plot of the Raman shift of 2D band vs G band for each of the 50 different spectra ($\lambda_{exc} = 532$ nm) of SWNT (black rhombi), MINT-exTTF (yellow circles), MINT-pyr (purple squares), and MINT-AQ (cyan triangles), data points are shaded to indicate the frequency of occurrence

Methods and Supplementary Figs. 1-4). For the synthesis of MINT-AQ, we used a clipping strategy in which the SWNTs serve as template for the formation of mac-AQ around the nanotubes, to yield MINT-AQ. Analysis of the kinetics of formation of MINT-AQ confirms that ring-closing metathesis around the SWNTs is the major reaction pathway, with negligible participation of oligomerization. Control experiments using $C_{60}$ as soluble template for RCM are also in agreement with this picture (Supplementary Fig. 8).

After the MINT-forming reaction, the SWNTs showed a loading of macrocycle of 35–37% for MINT-exTTF, 24–28% for MINT-pyr, and 31–33% for MINT-AQ by TGA analysis (Supplementary Fig. 5). This degree of functionalization remains stable even after reflux in tetrachloroethane for 30 min, which demonstrates the extreme stability of MINTs.

A preliminary TEM study performed in conventional equipment at 200 kV allows for the visualization of contrasts of adequate size and shape to be individual macrocycles around the SWNTs (or their decomposition products after reaction with the electron beam)[24]. Fig. 1c, d show representative examples of conventional TEM images of isolated SWNTs (diameter 0.8–0.9 nm) around which we can observe circular objects of ca. 2.0 nm diameter, marked at the images. In order to get more precise structural information, atomically resolved images were acquired in an aberration-corrected microscope at low voltage, 60 kV, in order to minimize the electron beam damage. A characteristic aberration-corrected HR-TEM image is shown in Fig. 1e. The image shows a macrocycle of diameter 1.7–1.8 nm around a SWNT of 0.9 nm in diameter. Remarkably, the distances between mac-AQ and the walls of the SWNT (see contrast profile in Fig. 1f) correspond to nearly ideal (0.35 nm, top) or very close (0.42 nm, bottom) van der Waals contacts, indicating a very strong interaction between mac-AQ and the SWNT, which justifies the template effect during the synthesis of MINT-AQ. The experimental distances observed in between the dark contrasts that compose the macrocycle are around 0.12 nm, in agreement with the average theoretical carbon–carbon distances.

The AFM images obtained upon exploration of a drop-casted suspension of MINT-AQ on mica are also consistent with the proposed rotaxane-like structure. Figure 1g, h show AFM topographic images of individualized SWNTs of height around

0.6–1.0 nm, which show protuberances of approximately 2 nm height (see profiles in Fig. 1i, j). In the phase images (Fig. 1k, l), these objects show different contrast compared to the SWNTs, demonstrating that they are not carbon nanotube protrusions or deformations.

We have previously shown that mac-exTTF behaves as an electron donor toward (6,5)-SWNTs[13]. The steady-state photophysical characterization of MINT-AQ (UV-vis-NIR and PLE) confirms that mac-AQ acts as an electron acceptor toward SWNTs, at least upon photoexcitation (Supplementary Figs. 6 and 7). Raman spectroscopy is particularly useful in characterizing the electronic properties of SWNTs. Figure 2a displays the average of 50 Raman spectra ($\lambda_{exc} = 532$) of MINT-exTTF (yellow), MINT-pyr (purple), MINT-AQ (cyan), and pristine (6,5)-SWNT (black). All spectra are very similar, with no increase in the relative intensity of the D band, proving that the covalent structure of the SWNT is preserved upon formation of MINTs. Analysis of the Raman shifts of the G and 2D bands is usually considered the best indication of doping in SWNTs. For direct and strong doping of SWNTs via electrochemical or electronic means, the expected direction and magnitude: large (up to 10 cm$^{-1}$) blue shift for p-doping, moderate (<5 cm$^{-1}$) red shift for n-doping, of the Raman shifts for the G band is well established[25]. Unfortunately, the case is not so clear for molecular dopants, since the Raman shifts are affected by other factors such as aggregation or mechanical strain that can change during the chemical treatment, and moderate blue shifts have been related to an increase in conductivity that can be found for both types of dopants[26]. In our case, both MINT-exTTF (1589 ± 2 and 2616 ± 4 cm$^{-1}$) and MINT-AQ (1589 ± 1 and 2619 ± 2 cm$^{-1}$) showed small hypsochromic shifts in the frequency of the G and 2D bands with regards to pristine SWNTs (1587 ± 2 and 2612 ± 5 cm$^{-1}$), while MINT-pyr shows smaller bathochromic shifts (1585 ± 2 and 2611 ± 3 cm$^{-1}$). Recently, Ryu and coworkers have shown that the mechanical strain and charge-doping components of the Raman shifts can be separated for graphene by plotting the Raman shift of the G band against that of the 2D band, whereby variations due to mechanical strain fall along a straight line of known slope while effects due to doping deviate from this behavior[27]. Since the origin of the G and 2D bands in SWNTs is identical to that of graphene[28], we reasoned

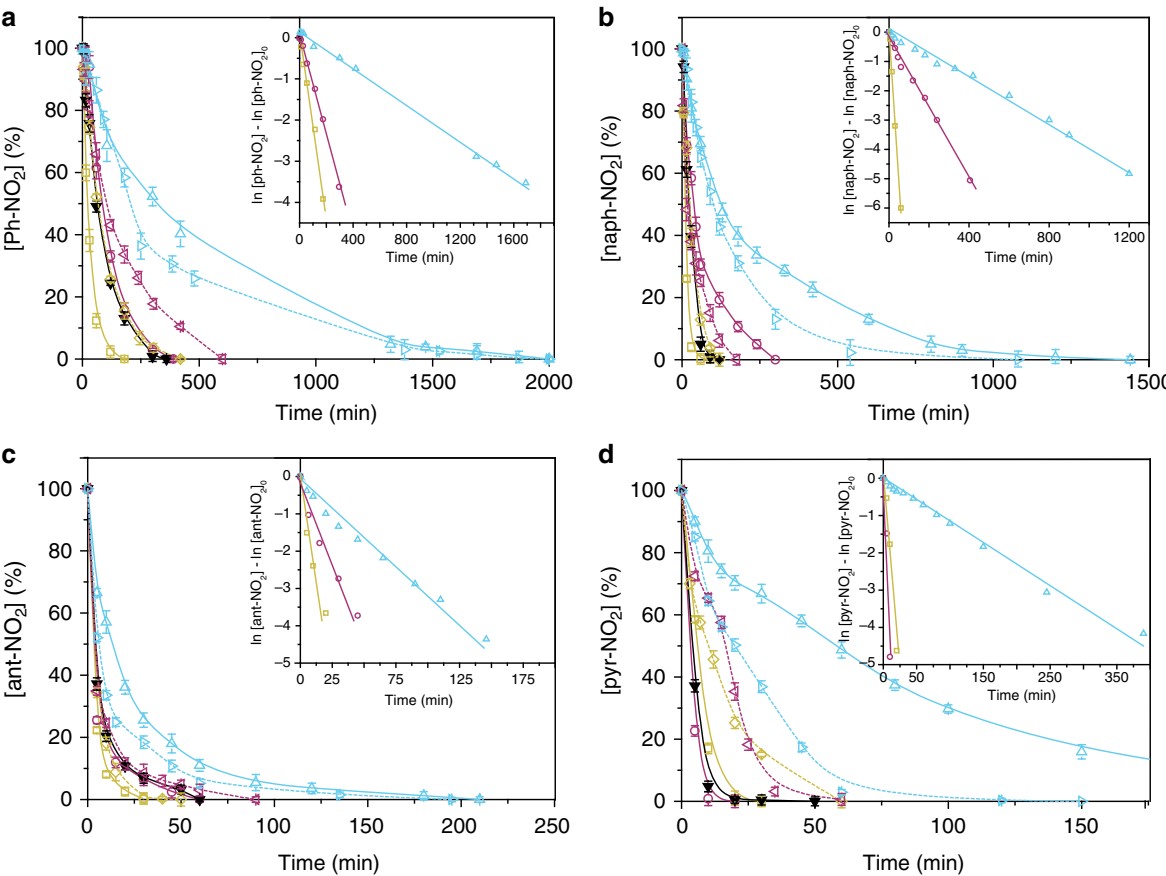

**Fig. 3** Catalysts' activity. Kinetics of the reduction of **a** Ph-NO$_2$, **b** naph-NO$_2$, **c** ant-NO$_2$, **d** pyr-NO$_2$ catalyzed by MINT-exTTF (yellow solid line), MINT-pyr (purple solid line), MINT-AQ (cyan solid line), 6,5-SWNT (black solid line), SWNT-exTTF (yellow dashed line), SWNT-pyr (purple dashed line), and SWNT-AQ (cyan dashed line). Error bars are standard deviation from three separate experiments

that a similar analysis would help us shed light on the origin of the Raman shifts. Figure 2b shows the corresponding plot. For SWNTs (black) and MINT-pyr (purple), the data show nearly identical linear tendencies, following a straight line of slope 1.2. In contrast, the data for MINT-AQ (cyan) shows a significantly larger slope of 1.65, while the MINT-exTTF data are closely grouped and show a significantly smaller (0.95) slope. By comparison with graphene[27], both MINT-exTTF and MINT-AQ data fall into the doping-affected quadrant, while MINT-pyr does not. Taken together, the Raman data are a solid experimental indication of the electronic effects of each type of macrocycle on the SWNTs: weak n-doping for mac-exTTF, weak p-doping for mac-AQ, no doping for mac-pyr, supporting the initial design. An analysis of Mülliken population confirms that there is a charge transfer between the macrocycle and SWNTs in the expected directions. For MINT-AQ, the charge transfer is from SWNT to mac-AQ leaving the SWNT with +0.011e (p-doping), while for the formation of the MINT-exTTF the charge transfer takes place from mac-exTTF to SWNT, leaving the latter with −0.043e (n-doping). This is consistent with the nature of electron-accepting tendency of AQ and electron-donor tendency of exTTF. For the MINT-pyr, the calculated charge transfer is +0.005e, one order of magnitude lower, and cannot be unambiguously identified as a charge transfer between the macrocycle and the SWNT. Furthermore, analysis of the localization and energy of the frontier molecular orbitals of energy-minimized (DFT, b97d/3-21g*) MINT models sufficiently large to reproduce the Raman results supports the predicted

electronic effects (Supplementary Figs. 10 and 11). These theoretical results are in good agreement with the experimental Raman tendencies in Fig. 2b.

**Catalysts activity and recyclability**. With the different influence of the macrocycles on the electronic properties of the MINT samples established, we went on to test its effects on the catalytic reduction of nitroarenes. Supramolecular complexes resulting from the direct mixing of SWNTs and mac-exTTF, mac-pyr, or mac-AQ were used as reference samples. A series of aromatic compounds with increasing conjugated aromatic rings: nitrobenzene (Ph-NO$_2$), 1-nitronaphthalene (naph-NO$_2$), 9-nitroanthracene (ant-NO$_2$), and 1-nitropyrene (pyr-NO$_2$), were selected as substrates to be reduced. Very briefly, the SWNT catalyst (5 mg) was mixed with 4.1 mmol of substrate in 2 mL of d$_6$-DMSO as solvent (selected after optimization of the reaction conditions) (Supplementary Fig. 13). Finally, 31.8 mmol of hydrazine were added as reducing agent, and the progress of the reaction was monitored by $^1$H-nuclear magnetic resonance (NMR) spectroscopy. In order to confirm that the active sites corresponded only to the nanotube walls, we conducted total reflection X-ray fluorescence (TRXF) measurements, and only ppm-level of metallic impurities were detected (Supplementary Figs. 15-18). More importantly, we used the same batch of SWNTs for all experiments and the synthesis of all MINTs, so any changes in catalytic activity can only be ascribed to modifications in the SWNT material. The reactions proceed smoothly with no induction period detected, yielding the aniline

**Table 1 First-order kinetic constants ($k$ values in s$^{-1}$ × 10$^{-3}$) as obtained from the fits shown in the insets of Fig. 3**

|              | Ph-NO$_2$ | naph-NO$_2$ | ant-NO$_2$ | pyr-NO$_2$ |
|--------------|-----------|-------------|------------|------------|
| MINT-exTTF   | 0.36      | 1.71        | 2.93       | 3.98       |
| MINT-pyr     | 0.20      | 0.36        | 1.28       | 7.98       |
| MINT-AQ      | 0.03      | 0.06        | 0.47       | 0.18       |
| (6,5)-SWNT   | 0.27      | 0.88        | 1.27       | 4.67       |
| SWNT-exTTF   | 0.19      | 0.60        | 2.83       | 1.10       |
| SWNT-pyr     | 0.07      | 0.45        | 0.86       | 3.98       |
| SWNT-AQ      | 0.08      | 0.12        | 0.48       | 0.72       |

reduction products almost exclusively, with the corresponding hydroxylamines as only detectable intermediates (Supplementary Fig. 14).

In Fig. 3, we compare the activity of all catalysts under study. In the absence of catalyst, the reduction of Ph-NO$_2$ proceeded to a conversion of only ca. 50% after 24 h of reaction (Supplementary Fig. 14). The first observation is that conversions >95% with selectivity >95% to the target aniline were achieved in all catalyzed reactions. Some noticeable tendencies are clear across the different substrates. For example, for Ph-NO$_2$, MINT-exTTF converted 95% of the starting material in 120 ± 5 min, while MINT-pyr needed more than twice as much time to reach the same conversion (300 ± 6 min); finally the reaction catalyzed by MINT-AQ required over 1800 ± 20 min to achieve 95% conversion. For comparison, the pristine (6,5)-SWNT showed an intermediate catalytic activity and required 260 ± 7 min to consume 95% of the starting material. This last observation confirms that the possible decrease in available SWNT catalytic sites due to the encapsulation with the macrocycles does not result in a significant decrease in the catalytic activity. An electronic effect, however, is clear. Larsen et al. showed that the carbon nanomaterial extracts electrons from the medium, stocks them as an electronic reservoir, and provides them for the reacting molecules[17]. According to this picture, the electron–donor behavior of mac-exTTF moiety should be beneficial, either by supplying additional electrons to the nanotube or, most likely, by decreasing the energy barrier toward release of the electrons. Following the same argument, the electronically "neutral" character of the mac-pyr should not influence the reactivity, and the electron-acceptor nature of mac-AQ should withdraw electron density from the nanotube, which would result in decreased activity. This is exactly the picture that emerges for the reduction of Ph-NO$_2$ and is conserved for all other substrates, except the largest pyr-NO$_2$, where the catalytic activity of (6,5)-SWNT, MINT-exTTF, and MINT-pyr is approximately equal. The detrimental effect of the electron-withdrawing mac-AQ is patent in all cases. Interestingly, the same relative tendencies are qualitatively reproduced in the supramolecular control experiments (mac-exTTF > mac-pyr > mac-AQ), but all supramolecular models show lower catalytic activity than the pristine SWNTs, except in the case of mac-exTTF for the reduction of ant-NO$_2$ (dashed lines, Fig. 3). It has been argued that residual carbonyl groups are responsible for the catalytic effect of SWNTs in the reduction of nitroaromatics[29]; our results contradict such hypothesis and support the picture provided by Larsen et al.[17].

The effect of the substrate on the catalytic reduction was also investigated. The reaction becomes progressively faster as the size of the aromatic nitroarene increases. For naph-NO$_2$, we needed 45 ± 2, 260 ± 4, and 1080 ± 10 min to achieve complete conversion using MINT-exTTF, MINT-pyr, and MINT-AQ, respectively.

Shorter times were required to reduce ant-NO$_2$ (20 ± 4, 45 ± 5, and 120 ± 3 min) and even shorter for pyr-NO$_2$ (16 ± 1, 8 ± 1 and 240 ± 5 min). DFT calculations were performed to evaluate the binding energy between the nitroaromatic molecules and the nanotube walls[34,35]. The calculations revealed a progressively more favorable binding energy between the nitroaromatic molecule and the carbon nanomaterial: −15.45 Kcal mol$^{-1}$ for Ph-NO$_2$, −21.73 Kcal mol$^{-1}$ for naph-NO$_2$, −26.54 Kcal mol$^{-1}$ for ant-NO$_2$, and −31.38 Kcal mol$^{-1}$ in the case of pyr-NO$_2$, as expected due to the increase in available surface of the nitroaromatic system. Therefore, the increase in reduction rate is directly related to an increase in SWNT–substrate binding energy.

Analysis of the first-order kinetic constants (insets of Fig. 3, and Table 1) allows for a more quantitative comparison of the results. MINT-exTTF is consistently the best catalyst and $k$ progressively increases with the number of condensed rings in the substrate. Meanwhile, the constants of the MINT-pyr are very similar to those obtained for the pristine (6,5)-SWNT and consistently smaller than those obtained for MINT-exTTF, with the exception of the pyr-NO$_2$ substrate. We interpret this exception in light of the strong tendency of pyrene to self-associate and the presence of two pyrene moieties in mac-pyr, which probably facilitates the adsorption of pyr-NO$_2$ on the catalyst surface. Finally, MINT-AQ consistently shows significantly smaller $k$ in all cases, reflecting the electron-acceptor character of mac-AQ.

Supramolecular control samples show the same tendencies and comparable but slower reaction rates for the first round of reactions, ruling out a direct catalytic effect of the macrocycle (Fig. 3). Remarkably, the supramolecular catalysts cannot be recycled, as they recover the basal activity of pristine (6, 5)-SWNTs upon purification after the first reaction cycle (Supplementary Figs. 26 and 27). In comparison, recycling of the MINT-based catalysts MINT-exTTF, MINT-pyr, and MINT-AQ, without any detectable loss in the macrocycle effects for up to 12 cycles was straightforward. In line with this recyclability results, the structural integrity of the MINT catalysts under the reaction conditions was probed by Raman spectroscopy (Supplementary Fig. 28).

## Discussion

In conclusion, we have shown that encapsulation within p- or n-doping macrocycles is a valid strategy for the regulation of the catalytic activity of SWNTs. As a test bed, we have chosen the reduction of nitroaromatic compounds. The effect of the macrocycles on the catalytic activity is most likely due to a combination of factors, including changes in the degree of aggregation, binding site availability, etc., but the electronic effect is clearly predominant. Electron-donating exTTF macrocycles lead to a higher activity, while electron-accepting mac-AQ moieties significantly slow the reaction rates. Meanwhile, SWNTs modified with the electronically neutral mac-pyr show very similar activity to pristine SWNTs. Crucially, mechanical interlocking of the macrocycles around the SWNTs to form MINTs results in stable catalysts that can be recycled, as opposed to classic supramolecular model compounds, which lose the effect of the macrocycle after the first reaction cycle. To achieve these conclusions, we purposely designed, synthesized, and fully characterized MINTs based on mac-AQ. We have also carried out DFT calculations on all MINTs, which support the picture provided by the experimental data.

This new artificial regulation strategy presents a combination of the features found in the natural regulation of enzymes that make it unique: it can be used for both positive and negative regulation, the effector (macrocycle) is associated with the catalyst (SWNT) via noncovalent yet stable mechanical bonds, and its

effect is remote but not allosteric, since it does not affect the 3D structure of the catalyst active site. We have focused here on the regulation of activity of SWNT catalysts, but we anticipate that structural variations on the macrocycles could make this strategy of general interest to help overcome the problems of aggregation and substrate specificity in one-dimensional catalysts.

## Methods

**Materials**. (6,5)-SWNTs were purchased from Sigma-Aldrich (0.7−0.9 nm in diameter, length ≥700 nm, mostly semiconducting, 95% purity). Reagents were used as purchased. All solvents were dried according to standard procedures. All air-sensitive reactions were carried out under $N_2$ atmosphere.

**Characterization**. Analytical thin-layer chromatographies were performed using aluminium-coated Merck Kieselgel 60 F254 plates. NMR spectra were recorded on a Bruker Avance 400 ($^1$H: 400 MHz; $^{13}$C: 100 MHz) spectrometers at 298 K, using partially deuterated solvents as internal standards. Coupling constants ($J$) are denoted in Hz and chemical shifts ($\delta$) in ppm. Electrospray ionization mass spectrometry and matrix-assisted laser desorption ionization (coupled to a time-of-flight analyzer) experiments were recorded on a HP1100MSD spectrometer and a Bruker REFLEX. TGA were performed using a TA Instruments TGAQ500 with a ramp of $10\,°C\,min^{-1}$ under air from 100 to $1000\,°C$. TEM images were obtained with JEOL-JEM 2100F instrument or a JEOL-JEM GRAND ARM300cF (AC-HRTEM). AFM images were acquired using a JPK NanoWizard II AFM working in dynamic mode. NT-MDT NSG01 silicon cantilevers, with typical values of 5.1 N $m^{-1}$ spring constant and 150 kHz resonant frequency, were employed under ambient conditions in air. TRXF analyses were performed on a TXRF 8030c - FEI Spectrometer. Raman spectra were acquired with a Bruker Senterra confocal Raman microscope instrument equipped with 532, 633, and 785 nm excitation lasers. UV-vis-NIR spectra were performed using a Shimadzu UV-VIS-NIR Spectrophotometer UV-3600. PLE intensity maps were obtained with NanoLog 4 HORIBA instrument.

**DFT calculations**. All theoretical DFT calculations were carried out within the DFT approach by using the C.01 revision of the Gaussian 09 program package[30]. Optimization and molecular orbitals calculations of MINT derivative were performed using the long-range corrected B97D density functional[31], which are able to incorporate the dispersion effects by means of a pair-wise London-type potential. The B97D density functional has emerged as a robust and powerful density functional able to provide accurate structures in large supramolecular aggregates dominated by non-covalent interactions of different nature. Raman spectra and analysis of Mülliken population were simulated by using the Coulomb-attenuated hybrid exchange-correlation functional (CAM-B3LYP) functional. This functional was developed by Yanai et al.[32], which includes the Hartree–Fock and the Becke exchanges as a variable ratio depending of the intermolecular distance. Both functional were combined with the Pople's 3-21G* basis set[33].

**Data availability**. All experimental data are available upon request.

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

## Acknowledgements

Funding from the European Union (ERC-Starting Grant: 307609 (to E.M.P.)), MINECO (Grants: CTQ2014-60541-P (to E.M.P.), JdC-2015-23531 postdoctoral fellowship (to B.N.-O.)), and the Comunidad de Madrid (Grant: MAD2D-CM program S2013/MIT-3007 (to E.M.P.)) is gratefully acknowledged. IMDEA Nanociencia

acknowledges support from the "Severo Ochoa" Programme for Centres of Excellence in R&D (MINECO, Grant SEV-2016-0686). The computational work was supported by the Campus de Excelencia Internacional UAM+CSIC. Additionally, we express our gratitude to the Supercomputing and Bioinnovation Center (SCBI) of the University of Málaga (Spain) for their support and resources. We thank the National Centre for Electron Microscopy (ICTS-CNME, Universidad Complutense) for electron microscopic facilities.

## Author contributions

M.B. designed and performed the catalysis experiments and analyzed the kinetic data. B. N.-O. performed the DFT calculations. A.d.J., M.V., and A.L.-M. synthesized and characterized materials. S.C. performed the AFM measurements. J.M.G.-C., H.S., and L. R.G. performed the ac-HRTEM measurements. E.M.P. designed and supervised research and wrote the manuscript with contributions from all authors.

## Additional information

**Competing interests:** The authors declare no competing interests.



