## [Peer Review File · Nature Communications]

Reviewers' comments:

Reviewer #1 (Remarks to the Author):

This manuscript reports about the modulation of the catalytic action of carbon nanotubes by controlled encapsulation with macrocycles which act as electron donor/electron acceptors. Compared to other approaches where carbon atoms of the nanotube are substituted with heteroatoms (e.g., B, N), this is an appealing alternative, which also allows to contrast bundling issues. I think that the results presented by the authors are significant and interesting for the broad audience, given the relevance of carbon nanotubes in materials science and nanotechnology. For this reason, I recommend the paper for publication after addressing a few issues listed below.

1. As discussed by the authors, the basic mechanism responsible for the modulation of the catalytic activity is the p/n doping of the nanotubes, i.e. charge transfer from the encapsulating macrocycles to the tube. Unfortunately, from the DFT calculations reported in the manuscript, one cannot easily get information about the net charge transfer from the macrocycles to the nanotube model (the orbitals in fig. S11 are not suitable and are hard to see in detail). However, since the electron density is at the core of any DFT calculation, this information should be readily available, and should be provided to the reader as an indication of the magnitude of the charge transfer.

2. The net charge transfer obtained by DFT should be related with the Raman data trends reported in Figure 2B.

3. The selected small basis set for DFT calculations (3-21G*) is fully understandable to keep under control the computational burden, but it could lead to a poor description of the electron density. I wonder if moderately more accurate basis sets, such as 6-31G(d) or 6-31G(d,p) would be feasible with present technology. In the case of one of the macrocycles (a smaller model than the encapsulated nanotube), it could be informative to the reader to see a comparison of the performance of 3-21G* vs. higher basis sets, so to assess any systematic deviation in the prediction of the Raman spectra (e.g., position of the main Raman lines). Since this kind of numerical assessments could be fairly time consuming, and incompatible with Nature Communication workflow, I leave to the Editor and to the Authors the final word on this point.

4. The binding energies between the macrocycles and the nanotube model are computed without taking into account the basis set superposition error (BSSE), which is known to be basis-set dependent, and more severe in the case of small basis sets. I suggest to carry out BSSE-corrected calculations of the binding energy of selected cases, to check the magnitude of the error, and determine whether it affects the conclusions reached based on the data presented in Tables S1,S2,S3.

Reviewer #2 (Remarks to the Author):

This ms reports on an elegant work aiming at modifying the activity of SWCNTs through encapsulation within electron donor/acceptor macrocycles.

The results sound well, and show a nice example of the influence of electron transfer in catalysis.

I suggest publication.

Reviewer #3 (Remarks to the Author):

Blanco and coworkers investigated the regulation of catalytic activity of carbon nanotubes through their encapsulation with macrocycles by forming pseudorotaxane like structures. The encapsulation of carbon nanotubes within the macrocycles demonstrated nicely by TEM measurements. Interestingly, by altering the electronic nature of the macrocycle, the authors were able to regulate the rate of the reaction for the reduction of nitrobenzene.

One question, which is not clear for this reviewer, is the nature of catalytic sites for the reduction reaction. It is clear that n-type doping of CNT by using extended-TTF leads to the enhanced catalytic activity, whereas the p-type doping by mac-AQ led to a slower reaction kinetics compared to CNT itself. This result, however, contradicts the recent article by Su et al. (Catal. Sci. Technol., 2014, 4, 1730). In this study, the authors observed enhanced catalytic activity for the phenanthraquinone functionalized CNTs (also noncovalent functionalization) when compared to CNT itself. And the authors established a linear relationship between the concentration of C=O moieties and the resulting catalytic activity. However, in the present study, the authors ruled out the participation of macrocycles in the catalytic reaction. Please explain?

FT-IR analysis (or in-situ FT-IR during the reduction reaction) should be carried out following each catalytic cycle in order to probe the chemical composition of macrocycles.

The authors should also carry out control experiments using physically mixed CNTs with the macrocycles, this particular experiment will be crucial to reveal the impact of pseudorotaxane formation on the catalytic activity as opposed to simple supramolecular functionalization.

The authors should also clarify the existence of any defect sites on the surface of CNTs, which could possibly serve as a catalytic site.

Another interesting experiment would be to oxidize exTTF macrocycle or reduce mac-AQ on the CNT surface. Provided that the origin of catalytic activity is solely based on CNT, this should reverse the effect of macrocycles.

Reviewer #1 (Remarks to the Author):

This manuscript reports about the modulation of the catalytic action of carbon nanotubes by controlled encapsulation with macrocycles which act as electron donor/electron acceptors. Compared to other approaches where carbon atoms of the nanotube are substituted with heteroatoms (e.g., B, N), this is an appealing alternative, which also allows to contrast bundling issues. I think that the results presented by the authors are significant and interesting for the broad audience, given the relevance of carbon nanotubes in materials science and nanotechnology. For this reason, I recommend the paper for publication after addressing a few issues listed below.

1. As discussed by the authors, the basic mechanism responsible for the modulation of the catalytic activity is the p/n doping of the nanotubes, i.e. charge transfer from the encapsulating macrocycles to the tube. Unfortunately, from the DFT calculations reported in the manuscript, one cannot easily get information about the net charge transfer from the macrocycles to the nanotube model (the orbitals in fig. S11 are not suitable and are hard to see in detail). However, since the electron density is at the core of any DFT calculation, this information should be readily available, and should be provided to the reader as an indication of the magnitude of the charge transfer.

2. The net charge transfer obtained by DFT should be related with the Raman data trends reported in Figure 2B.

We are really thankful to Referee#1 for these two comments. This is clearly a very important matter that we had failed to consider in our original manuscript. We have now included Mülliken population analyses in the main text and the SI. The results are in accordance with the p/n doping of the nanotubes upon formation of MINTs with the different macrocycles (p-doping for AQ, n-doping for exTTF, negligible doping for pyr). They are also in good agreement with the experimental Raman results of figure 2B. Besides this, we have re-designed Figure S11 to make the differences in the frontier MOs more clear.

3. The selected small basis set for DFT calculations (3-21G*) is fully understandable to keep under control the computational burden, but it could lead to a poor description of the electron density. I wonder if moderately more accurate basis sets, such as 6-31G(d) or 6-31G(d,p) would be feasible with present technology. In the case of one of the macrocycles (a smaller model than the encapsulated nanotube), it could be informative to the reader to see a comparison of the performance of 3-21G* vs. higher basis sets, so to assess any systematic deviation in the prediction of the Raman spectra (e.g., position of the main Raman lines). Since this kind of numerical assessments could be fairly time consuming, and incompatible with Nature Communication workflow, I leave to the Editor and to the Authors the final word on this point.

We agree with Referee#1, a higher basis set would have been desirable, especially for Raman calculations, as it is usual practice for conjugated molecules. Unfortunately, the size of the systems (between 282 and 306 atoms, depending on the MINT) and, mainly, the resonance Raman effect of SWCNTs, complicate the Raman calculations. In Table S4, we now include a comparative between both basis sets: 6-31g* and 3-21g*, and the tendencies in the charge-

transfer results do not change between the two basis sets, although they do seem quantitatively larger for 6-31g*. Unfortunately, to obtain Raman spectra at a higher level of theory is difficult in these systems. We have tried, but we did not succeed due to computing time issues. In that sense, we decided to show all the results using the same basis set, to avoid complicating the reading of the manuscript for the broader audience.

4. The binding energies between the macrocycles and the nanotube model are computed without taking into account the basis set superposition error (BSSE), which is known to be basis-set dependent, and more severe in the case of small basis sets. I suggest to carry out BSSE-corrected calculations of the binding energy of selected cases, to check the magnitude of the error, and determine whether it affects the conclusions reached based on the data presented in Tables S1,S2,S3.

We thank the referee for this comment. All binding energies in our original submission were already corrected using the basis set superposition error (BSSE) according to the counterpoise (CP) scheme following the work of Boys et al. (Mol. Phys. 1970, 19, 553), but we did not mention it. We have now included a sentence in the SI to correct this mistake. Additionally, we would like to mention that the computed binding energies in this work are similar to related systems we have studied earlier (see: Chem. Eur. J. 2017, 23, 1290, Chem. Sci., 2015, 6, 7008).

Reviewer #2 (Remarks to the Author):

This ms reports on an elegant work aiming at modifying the activity of SWCNTs through encapsulation within electron donor/acceptor macrocycles.

The results sound well, and show a nice example of the influence of electron transfer in catalysis.

I suggest publication.

We thank Reviewer #2 for his/her positive comments.

Reviewer #3 (Remarks to the Author):

Blanco and coworkers investigated the regulation of catalytic activity of carbon nanotubes through their encapsulation with macrocycles by forming pseudorotaxane like structures. The encapsulation of carbon nanotubes within the macrocycles demonstrated nicely by TEM measurements. Interestingly, by altering the electronic nature of the macrocycle, the authors were able to regulate the rate of the reaction for the reduction of nitrobenzene.

We are glad that Reviewer #3 found our results interesting.

One question, which is not clear for this reviewer, is the nature of catalytic sites for the reduction reaction. It is clear that n-type doping of CNT by using extended-TTF leads to the enhanced catalytic activity, whereas the p-type doping by mac-AQ led to a slower reaction kinetics compared to CNT itself. This result, however, contradicts the recent article by Su et al. (Catal. Sci. Technol., 2014, 4, 1730). In this study, the authors observed enhanced catalytic activity for the phenanthraquinone functionalized CNTs (also noncovalent

functionalization) when compared to CNT itself. And the authors established a linear relationship between the concentration of C=O moieties and the resulting catalytic activity. However, in the present study, the authors ruled out the participation of macrocycles in the catalytic reaction. Please explain?

The discrepancy of our results with those described by Su et al. in *Catal. Sci. Technol.* (reference 28 in the original manuscript) is clear. As we already commented in our original submission, all the data reported in our manuscript (experiments and theory) are in line with the catalytic effect depending on the electronic density of the SWCNT surface, which agrees with the postulation of the reaction mechanism by Larsen et al. in *Carbon* (ref 16 of the manuscript). Please note that, in our case, the commercially available SWCNTs are quasi-defectless (see Raman data), (6,5)-enriched SWCNTs, with a carbon nanotube content > 99%. In comparison, the nanotubes synthesized by Su et al. are at least partially MWCNTs, with a significant level of defects, and significant amount of impurities, in particular Fe (see characterization data in reference 28).

FT-IR analysis (or in-situ FT-IR during the reduction reaction) should be carried out following each catalytic cycle in order to probe the chemical composition of macrocycles.

We thank Reviewer #3 for this suggestion. We were not able to detect the signals of the macrocycle in the FTIR of any catalyst sample, most likely due to the large absorption of (6,5)SWCNTs in the IR region. In the reaction mixture, this would be further complicated by the presence of the reagents. We therefore cannot monitor the structural integrity of the macrocycles during the reaction using FTIR. We believe the constant activity of the catalysts during several cycles (Figure S26) is quite compelling evidence of their structural integrity. We have now also carried out Raman before and after reaction for the MINT-AQ catalyst, which shows superimposable spectra (Figure S28).

The authors should also carry out control experiments using physically mixed CNTs with the macrocycles, this particular experiment will be crucial to reveal the impact of pseudorotaxane formation on the catalytic activity as opposed to simple supramolecular functionalization.

We do not fully understand this comment by Reviewer #3. Control experiments in which we mixed pristine nanotubes with the macrocycles had already been carried out. See figure 3, table 1, lines 201-206 and 242-244 of the main text in our original submission, and Figures S23-S25 and S27 in the SI, to see the modification of the performance with the supramolecular structures. In fact, the difference between these controls and our MINT samples is one of the main findings of our manuscript, as detailed in the conclusions: “Crucially, mechanical interlocking of the macrocycles around the SWNTs to form MINTs results in stable catalysts that can be recycled, as opposed to classic supramolecular model compounds, which lose the effect of the macrocycle after the first reaction cycle.”

The authors should also clarify the existence of any defect sites on the surface of CNTs, which could possibly serve as a catalytic site.

As described by our supplier (Sigma Aldrich, CAS Number 308068-56-6) and found experimentally by us using Raman (Figure 2 and Figure S10) and confirmed by TEM imaging) the nanotubes employed for these experiments are quasi-defectless. We believe, based on these data and the mechanism proposed by Larsen (ref. 16), that the defects are not the active site of the reduction. To further confirm this, Raman spectroscopy was also performed after the catalytic runs in order to detect any possible modification of the carbon structure after reaction. No modification was observed. Results have been included as new Figure S28 and mentioned in the main text.

Another interesting experiment would be to oxidize exTTF macrocycle or reduce mac-AQ on the CNT surface. Provided that the origin of catalytic activity is solely based on CNT, this should reverse the effect of macrocycles.

We consider this as a very interesting idea. However, it is at least difficult to carry it out experimentally. Finding the optimized conditions to reduce/oxidize in situ all the macrocycles without altering the SWCNT structure, characterizing the new systems, and making the reducing conditions of the reactions compatible with the oxidized macrocycles seems rather challenging. We will consider it for future studies.

REVIEWERS' COMMENTS:

Reviewer #1 (Remarks to the Author):

I have examined the revised version of the manuscript and considered the replies by the authors in the rebuttal letter. I am convinced of the soundness of the work. I therefore recommend its publication.

Reviewer #3 (Remarks to the Author):

Considering the clarifications/improvements made by the authors, the manuscript is now suitable for publication. This reviewer also notes the different nanotube used by Sun and coworkers, which could explain the discrepancy in the catalytic activity. Overall, this is an interesting piece of research, and thus it should appeal to the broad readership of Nature Communications.